# Stock Status of Noncommercial Fish Species in Aras Dam Reservoir: Mismanagement Endangers Sustainable Fisheries

**DOI:** 10.3390/biology14091242

**Published:** 2025-09-11

**Authors:** Ali Haghi Vayghan, Mehrnaz Ghanbarzadeh, Nan-Jay Su

**Affiliations:** 1Department of Ecology and Aquatic Stocks Management, Artemia and Aquaculture Research Institute, Urmia University, Urmia 57561-15311, Iran; 2Department of Natural Resources and Environmental Engineering, School of Agriculture, Shiraz University, Shiraz 71441-13131, Iran; mehrnaz.ghanbarzadeh@saadi.shirazu.ac.ir; 3Department of Environmental Biology and Fisheries Science, National Taiwan Ocean University, Keelung 20224, Taiwan; 4Center of Excellence for the Oceans, National Taiwan Ocean University, Keelung 20224, Taiwan

**Keywords:** Aras Dam Reservoir, fishery management, sustainable fisheries, data-limited method, stock status

## Abstract

This study addressed the critical issue of unsustainable fisheries in the Aras Dam reservoir, a vital source of fish in northwest Iran, where fish populations are threatened by mismanagement and outdated information. Our goal was to assess the current health of four important noncommercial fish species: the silver bream, common bream, common roach, and freshwater bream. Using advanced computer models that can work with limited data, we analyzed historical catches and fish measurements. Our results showed that the silver bream population is healthy and could potentially support more fishing. However, the populations of common bream, common roach, and freshwater bream are overfished or severely depleted, meaning current fishing levels are not sustainable. These findings are essential for local fishery managers. They provide clear guidance to implement measures like limiting overall catches, reducing fishing pressure, and stopping illegal fishing. This will help these fish populations recover and ensure that the Aras Dam reservoir remains a valuable resource for local communities for generations to come.

## 1. Introduction

The number of overfished stocks has increased worldwide, leading to a decline in fish abundance and marine ecosystems [1]. Thus, stock management strategies should be applied to ensure sustainable fisheries worldwide. All stocks that are being exploited should be monitored and managed [2]. However, limited data and expertise for doing so render the monitoring and management of fish stocks challenging. In recent years, data-limited methods such as the catch maximum sustainable yield (CMSY) model have been used to estimate reference points for fisheries and provide reasonable predictions of relative biomass and exploitation rates [3,4]; such predictions can provide a comprehensive understanding of the stock status of fish species and the amount of overfishing by local fishermen.

Data-limited methods have been widely used, often in combination with other parallel methods, to evaluate the stocks of different fish species, including approximately 400 European stocks [5], >1300 exploited marine fish and invertebrate species [6,7,8,9,10]. However, the CMSY model is associated with uncertainty and errors that must be addressed [11,12]. Nevertheless, CMSY is one of the most rapid methods for evaluating the stock of data-poor fisheries, providing timely insights that can guide the formulation of policies for sustainable fishery management. While other rapid assessment approaches, such as depletion-based or length-based methods, are available, CMSY was chosen for this study due to its proven applicability in data-limited contexts and its suitability for the inland fish stocks of the Aras reservoir.

The Aras River is an internationally important river, with a length of 1072 km, average discharge of 5323 × 10^6^ m^3^/year, and abundance of biogenic compounds and exhibits seasonal temperature fluctuations and is one of the most important feeders of the Caspian Sea. It originates in Turkiye and crosses Armenia, Iran, and Azerbaijan, and it has varying hydro-political interactions across these regions and economic, social, and ecological consequences [13,14]. In northwest Iran, the Aras Dam reservoir is one of the main sources of inland fisheries. Notably, >3.7 × 10^3^ mt [15] of more than 12 of the 27 species in the Aras River ecosystem are harvested legally and illegally from the Aras Dam reservoir each year [16]. This reservoir is economically crucial to local fishermen. Notably, illegal fishing in the Aras Dam reservoir has increased, threatening local legal fishery companies by causing low fish production and high stock pressure; these threats have led to a decrease in the stocks of commercial species and an increase in the stocks of noncommercial species in the reservoir.

Exploited noncommercial fish species in the Aras Dam reservoir include silver bream (*Blicca bjoerkna*), common bream (*Abramis brama*), common roach *(Rutilus rutilus*), and freshwater bream (*Carassius auratus*), which are mainly used in fish powder factories. The four species selected account for over 90% of the total catches in the Aras Dam reservoir. Although they are mostly noncommercial, their dominance in terms of abundance and biomass makes them representative indicators for assessing the overall stock status of the reservoir. Notably, the most recent study regarding the fish fauna and stock status of Aras Dam reservoir was that by Abbasi and Sarpanah [16], who conducted research from 1994 to 1995. No contemporary study has investigated the stock status for this crucial ecosystem. Any threat to the fish stock in the Aras Dam reservoir due to mismanagement would have natural, economic, and social consequences.

## 2. Materials and Methods

### 2.1. Study Area and Fishery Data

The Aras Dam was established over the Aras River in 1970; it has a height of 40 m (130 ft). The Aras River forms a section of Iran’s international boundaries with Armenia, Azerbaijan, and Turkey, and it is one of the most crucial and largest rivers in northwest Iran (Figure 1). The river has a length of 1072 km and flows into the Caspian Sea after joining the Kura stream [17]. The Aras River has a unique ecosystem in terms of fish species diversity; 27 species have been detected in the river [16].

The current study obtained time-series catch and fishing effort data from Beach seins net fishing (950 m length, 18 m height, and 35 mm mesh size in the central part of the net) to calculate catch per unit effort (CPUE) from 2013 to 2022 that were collected from west Azerbaijan fisheries officials in Iran; these data were input into the CMSY^++^ model. Legal fishing by fishing companies is conducted from late September to early April because of the fishing ban policy of west Azerbaijan fisheries. In this study, illegal fishing activity was estimated using the daily fish input data of fish powder factories located in the Aras Dam area. Estimates of illegal catches derived from fish supplied to the factory were cross-checked against official fisheries landing reports and independent local market surveys.

### 2.2. CMSY^++^ and BSM Methods

CMSY^++^ is an advanced state-space Bayesian method for stock assessment that estimates reference points for fisheries (*MSY*, *F_MSY_*, and *B_MSY_*). It also estimates status or relative stock size (*B*/*B_MSY_*) and fishing pressure or exploitation (*F*/*F_MSY_*) from catch and (optionally) abundance data, a prior for resilience or productivity (*r*), and broad priors for the ratio of biomass to unfished biomass (*B*/*k*) at the beginning of the year, midyear, and the end of the time series. This model is a more advanced version of the Catch-MSY model developed by Martell and Froese [4]. The basis of both these models is the modified Schaefer surplus production model, which enables fitting of abundance indicators, if available.

The CMSY^++^ model is a further advancement of the CMSY method developed by Froese et al. [3]. CMSY^++^ overcomes several of the limitations of the CMSY method. A major improvement in the CMSY method and Bayesian state-space implementation of the Schaefer model (BSM) is the introduction of multivariate normal priors for *r* and *k* in log-space, replacing the previous uniform prior distributions. This enables a simplified means of determining the “best” *r*–*k* pair in CMSY and achieves faster run times. The CMSY^++^ model is based on the first derivative of the logistic curve of population growth; the numbers of individuals are replaced by the sum of their body weights [18]. The CMSY^++^ model is represented as Equation (1):(1)Bt+1={Bt+r1−Btk−Cteεt}eƞt
where *B_t_* is the biomass, *C_t_* is the catch in metric tons (t) in year *t*, *r* (year^−1^) is the intrinsic rate of population growth, and *k* is the carrying capacity of the environment for this population in metric tons (t). *ε_t_* is the normally distributed observation error of catches, and η*t* is the process error. They are implemented as lognormal error terms. These lognormal error terms are omitted in subsequent equations. Thus, if reasonable estimates of the start biomass and *k* are available for quantifying the unexploited and initial stock sizes and if a reasonable estimate of *r* can be inferred from life-history traits [12], a time series of biomass can be projected on the basis of the time series of catches, with the maximum sustainable fishing mortality calculated as *F_MSY_* = *r*/2 and the minimum biomass that can produce maximum sustainable yield (MSY) calculated as *B_MSY_* = *k*/2. This approach is referred to as stochastic reduction analysis [19,20].

If stock biomass is severely depleted (<0.25 *k*), recruitment is reduced. This is expressed by Equation (2), which is a slight modification of Equation (1):(2)$$B_{t+1}=B_{t}+\left(\frac{4B_{t}}{k}r\right)\left(1-\frac{B_{t}}{k}\right)B_{t}-C_{t}|\frac{B_{t}}{k}<0.25$$
where (4*B_t_*/*k*) results in a linear decline of *r* if the biomass is less than *k*/4, which accounts for reduced recruitment and thus decreased productivity at a low population size. Half of *B_MSY_*, which is *k*/4 in the Schaefer model, is typically employed as a proxy for the biomass threshold, below which recruitment may be reduced.

The Schaefer model can be expressed as a function of *r* and MSY, without *k*, as indicated in Equation (3). However, this does not change the dynamics of the model, and the surplus production seems to be less intuitive than that in Equation (1):(3)$$B_{t+1}=B_{t}+r\cdot B_{t}-\frac{{\left(rB_{t}\right)}^{2}}{4MSY}-C_{t}$$

To retain the original form of the CMSY base model, as expressed in Equation (2), with parameters *r* and *k*, the within-stock correlation between *r* and *k* is accounted for in a multivariate lognormal (MVLN) distribution. This distribution is implemented by drawing a large sample (*n* = 10,000) of independent log($\tilde{r}$) and log($\tilde{MSY}$) from their prior distributions, computing the corresponding log($\tilde{k}$) = log(4) + log($\tilde{MSY}$) − log($\tilde{r}$), and computing the means and covariance of log($\tilde{r}$) and log($\tilde{k}$), which are then passed on as covariance matrix for the *r*–*k* ~ MVLN prior in the CMSY^++^ model and BSM formulations.

A feed-forward artificial neural network (ANN) is selected for classifying the stock status as being above or below the MSY level, as expressed in Equation (4):(4)$$({B/k)}_{tprior}=\frac{1+A_{t\sqrt{1-C_{t}/{MSY}_{prior}}}}{2}$$

This equation only provides real-number solutions if *C_t_* ≤ *MSY_prior_*. Therefore, its application is restricted to cases where *C_t_* < 0.99 *MSY_prior_*.

This equation presents the derivation of a point estimate of relative equilibrium biomass (*B*/*k*) from catch data relative to MSY. Catch and biomass are rarely in equilibrium in real-world stocks, and the width and shape of uncertainty vary with the position of the equilibrium point estimate in the *B_t_*/*k*–*C_t_*/*MSY* space.

The equilibrium curve for the interplay between relative biomass (*B*/*k*) and relative catch (*C_t_*/*MSY*) in the modified Schaefer model is derived in Equation (5):(5)$$\frac{C}{MSY}=(4\frac{B}{k}-({2\frac{B}{k})}^{2})\cdot RC$$
where *RC* denotes recruitment correction; *RC* = 4 *B*/*k* if *B*/*k* < 0.25 and *RC* = 1 otherwise, as is true for same as in Equation (2).

The equilibrium curve for the model developed by Fox (1970) is derived from Equation (6):(6)$$\frac{C}{MSY}=e\frac{B}{k}(1-\log\left(e\cdot \frac{B}{k}\right))$$
where *e* represents Euler’s number 2.718.

For verification, the predictions of the CMSY^++^ method are compared with simulated data, where the “true” values of parameters and biomass data are known. For evaluation against real-world fishery data, the predictions of the CMSY^++^ method are compared against corresponding parameters and abundance estimates derived from fully or partly assessed stocks, for which biomass or CPUE data are available in addition to catch data. For this purpose, BSM is developed, where *r*, *k*, and *MSY* are predicted from catch and abundance data. The basic biomass dynamics are governed by Equation (7):(7)$$B_{t+1}=B_{t}+r(1-B_{t}/k)B_{t}-C_{t}$$
where *B_t_* is the biomass in year *t*, *B_t+_*_1_ is the exploited biomass in year *t* + 1, *r* is the intrinsic rate of population increase, *k* is the carrying capacity (i.e., mean unexploited stock size), and C*_t_* is the catch data in year *t*.

If stock biomass is severely depleted and is <0.25 *k*, recruitment is reduced. This is expressed by Equation (8), which is a slight modification of Equation (7):(8)$$B_{t+1}=B_{t}+\left(\frac{4B_{t}}{k}r\right)\left(1-\frac{B_{t}}{k}\right)B_{t}-C_{t}|\frac{B_{t}}{k}<0.25$$
where the term 4*rB_t_*/*k* denotes the assumption that the intrinsic rate of population growth declines linearly with biomass, which is less than half the biomass associated with MSY. Resilience data in FishBase and SeaLifeBase are translated into the prior of *r*-ranges.

The prior of *k* is computed using the following equations:(9)$$k_{low}=\frac{\max\left(C\right)}{r_{high}};\;k_{high}=\frac{4\max\left(C\right)}{r_{low}}$$(10)$$k_{low}=\frac{2\max\left(C\right)}{r_{high}};\;k_{high}=\frac{12\max\left(C\right)}{r_{low}}$$
where *r_low_* and *r_high_* are, respectively, the lower and higher bounds of the range of *r*-values explored using the CMSY method, *k_low_* and *k_high_* are, respectively, the lower and higher limits of *k*, and *max*(*C*) is the maximum catch in the available data.

The BSM method (i.e., including relative abundance data) is used to calculate the standard deviation of *r*, which falls between 0.001 and 0.02 *irf*.(11)$$irf=3/(r_{high}-r_{low})$$
where *irf* is a factor applied to estimate *r* and *r_high_* and *r_low_* are ranges suggested in FishBase (www.fishbase.org).

The Schaefer model is used to transform the CPUE into biomass on the basis of the catchability coefficient *q* and is presented in Equation (12):(12)$${CPUE}_{t}=q\cdot B_{t}$$
where *CPUE_t_* and *B_t_* are the mean CPUE and biomass in year *t*, respectively, and *q* is the catchability coefficient.

The population dynamic can be expressed by relative abundance (CPUE) using Equation (13):(13)$${CPUE}_{t}+1={CPUE}_{t}+r(1-{CPUE}_{t}/qk){CPUE}_{t}-qC_{t}$$
where the variables and parameters are defined as in Equations (7) and (12). The prior *q* is derived from Equation (14):(14)Y=rB(1−B/k)
where *Y* is the equilibrium yield for *B* and the other parameters are defined as in Equation (7).

BSM can be used to estimate *q*. For stocks with high recent biomass, the lower and higher priors for *q* are calculated using Equations (15) and (16) as follows:(15)$$q_{low}=0.25r_{pgm}{CPUE}_{mean}/C_{mean}$$(16)$$q_{high}=0.5r_{high}C{PUE}_{mean}/C_{mean}$$
where *q_low_* and *q_high_* are the lower and upper prior catchability coefficients for stocks with high recent biomass, respectively, *r_pgm_* is the geometric mean of *r*, *r_high_* is the upper prior range for *r*, *CPUE_mean_* is the mean CPUE over the last 5 or 10 years, and *C_mean_* is the mean catch over the same period.

*B_start_*/*k* and *B_end_*/*k* are the priors of relative biomass at the start and the end of each time series, and their ranges are estimated on the basis of the assumed depletion level [3].

All available *r* values are assigned to bins in log-space with equal widths; the most probable *r* value is derived from the 75th percentile of the mid-values of the occupied bins. If the *r* value is higher than the 50th percentile of the mid-values of the occupied bins, the most probable *k* value is derived through linear regression:(17)MSY=rk/4→log(k)=log(4MSY)+(−1)log(r)

The relative biomass (*B*/*B_MSY_*) in the final year, which represents the status of a stock, can be estimated using both CMSY and BSM. In addition, Kobe plots, which are generated using relative fishing mortality coefficients and the ratio of fishing mortality to maximum sustainable fishing mortality (*B*/*B_SMY_* and *F*/*F_MSY_*, respectively), are used to simultaneously assess the status of all stocks in the most recent year. To ensure consistency in interpretation, stock status categories were defined following Froese et al. [21]. Specifically, stocks with *B*/*B_MSY_* < 0.5 were classified as considerably depleted, 0.5 ≤ *B*/*B_MSY_* < 0.8 as moderately depleted, and *B*/*B_MSY_* ≥ 0.8 as healthy.

### 2.3. Biometric Data and Length–Weight Relationship

To determine the length–weight relationship, this study analyzed 1206 fish specimens, including 463 Silver bream, 391 Common roach, and 352 Freshwater bream, collected from fishing cooperative positions using Beach seins net fishing (950 m length, 18 m height, and 35 mm mesh size in the central part of the net) once a week for 6 months (October 2023–March 2024). This study did not investigate the length–weight relationship for Common bream because this species was not present in the beach sein net and, consequently, biometric data could not be obtained for the species. For each specimen, the total length (*L*) was measured to the nearest 0.1 cm, and total weight (*W*) was measured to the nearest 0.1 g by using a measuring board and a digital scale, respectively. The parameters of the length–weight relationship were calculated for noncommercial species by fitting the power function to the length and weight data and applying Equation (18).(18)$$W=a\cdot L^{b}$$
where *W* is the wet body weight (g), *TL* is the total length (cm), *a* is the intercept, and *b* is the allometric coefficient. Pauly’s *t* test [22] was used to determine if the slope of relationships was significantly different from the value of 3.

### 2.4. Statistical Analysis

Statistical analyses were performed using SPSS (version 26) with R Studio (version 1.1.446) software. The R studio software program, which implements the CMSY^++^ method [21], was applied to assess the status of fishery stocks for the four noncommercial species considered in this study in the Aras Dam reservoir. In addition, BSM, which is part of the CMSY^++^, was applied to account for variability in both population dynamics (process error) and measurement and sampling (observation error) [3,23,24]. At the end of the time series, CMSY^++^ provided routine reference points for fisheries to enable exploration of the stock status of selected species. All data files and R-code of the CMSY^++^ method are provided in the supplement in Froese et al. [21].

## 3. Results

### 3.1. CMSY^++^ Model and BSM

This study estimated the main parameters for obtaining reference points for the fisheries of noncommercial species in the Aras Dam reservoir (Table 1 and Table 2) by using the CMSY^++^ model and BSM. According to the results obtained using the CMSY^++^ model and BSM, the estimated value of *B*/*B_MSY_* for *B. bjoerkna* was >1.0, and the value of *F*/*F_MSY_* was <1.0 (Table 1), indicating the Aras Dam reservoir has a healthy stock of this species (Kobe plot, Figure 2A,B and Figure 3A,B). The estimates obtained using the two models indicated an increase in fishing activity from 2013 to 2022, and the catch amount for the aforementioned species was lower than the MSY in all years. Consequently, during these years, the size (biomass) of the stock increased (Figure 2C and Figure 3C). The obtained reference points for fisheries were used to estimate the MSY for *B. bjoerkna*, which was 769 mt when the CMSY^++^ model was applied and 750 mt when the BSM was; the catch amount never reached these points or surpassed the sustainable level. Hence, the catch amount for this species can be slightly increased.

For *A. brama*, the estimated values for *B*/*B_MSY_* and *F*/*F_MSY_* when the CMSY^++^ model was used were both <1.0. When the BSM was used, the *B*/*B_MSY_* value was <1.0, and the *F*/*F_MSY_* value was >1.0 (Table 2). According to the values calculated using the CMSY^++^ model, the stock is overfished (Kobe plot, Figure 4A,B). According to the values calculated using BSM, overfishing of this species occurs, and the stock is in an overfished state (Kobe plot, Figure 5A,B). A graph was created of the predicted total catch versus the MSY obtained using the two models; it indicated a decreasing trend of fishing from 2013 to 2022 (Figure 4 and Figure 5C). According to the reference points for fisheries obtained using the CMSY^++^ model, the MSY for *A. brama* was nearly 168 mt; in 2013 and 2014, the catch amount was higher than the MSY was. This led to a decrease in the stock size (biomass).

Fishing pressure in the years following 2014 led to the catch amount consistently being lower than the MSY, and the high pressure led to stock depletion (Figure 4C), with the stock for this species being overfished. According to the BSM, the estimated MSY was lower (137 mt), and the stock status was similar to those calculated using the CMSY^++^ model. However, the catch amount was higher than the MSY in 2013 and 2015. Fishing pressure in the years following 2015 led to the catch amount consistently being lower than the MSY, and the high pressure led to stock depletion (Figure 5C), indicating this stock has been overfished and is in an overfished state. Because of its stock status and the absence of this species in the beach sein net, biometrics was not calculated during the sampling season for this species.

For *R. rutilus*, the CMSY^++^ model results revealed that the biological reference point of *B*/*B_MSY_* was mostly <1.0, whereas *F*/*F_MSY_* was mostly >1.0. According to these values, this stock is overfished and in an overfished state (Kobe plot, Figure 6A,B). A graph of the predicted total catch versus the MSY, calculated using the CMSY^++^ model, indicates an increasing trend of fishing from 2013 to 2022. The catch amount was lower than the MSY until 2020, after which it was higher. This indicates that the stock size (biomass) decreased because exploitation of this species surpassed the sustainable level (Figure 6C). For *R. rutilus*, the MSY was estimated to be 724 mt when the CMSY^++^ model was used.

When the BSM was used, the *B*/*B_MSY_* and *F*/*F_MSY_* were both > 1.0. These values indicate the species is in an overfished state (Kobe plot, Figure 7A,B). A graph of the predicted total catch versus the MSY, as calculated using BSM, reveals an increasing trend of fishing from 2013 to 2022. However, up until 2018, the catch amount remained lower than the MSY. After 2018, the catch amount exceeded the MSY, which led to a slight decrease in the stock size (biomass; Figure 7C). When the BSM was used, the MSY was 717 mt. In the years after 2019, the catch amount exceeded the MSY, indicating the stock of this species was overfished and in an overfished state.

For *C. auratus*, when the CMSY^++^ model was used, the biological reference point of *B*/*B_MSY_* was >1.0, and that of *F*/*F_MSY_* was <1.0. These values indicate that exploitation of the stock of this species exceeded the limits of reference points in the Aras Dam reservoir (Kobe plot, Figure 8A,B). When the BSM was used, the *B*/*B_MSY_* and *F*/*F_MSY_* values were both <1.0, indicating the stock of this species is recovering (Kobe plot, Figure 9A,B). A graph of the predicted total catch versus MSY obtained using the two models indicated an increasing trend in total fishing from 2013 to 2018, followed by a decline in fishing in the subsequent years. In all years, the catch amount consistently remained lower than the MSY. Consequently, according to the values obtained using the CMSY^++^ model, the stock size (biomass) considerably increased (Figure 8C). According to the values obtained using the BSM, the stock size increased slightly (Figure 9C). When the CMSY^++^ model was used, the MSY for *C. auratus* was 1050 mt, and when the BSM was used, it was 770 mt. These values indicate an increase in the catch amount for this species in the Aras Dam reservoir.

### 3.2. Length–Weight Relationship

Figure 10 presents the results regarding the length–weight relationships. The coefficient of determination (R^2^) ranged between 0.783 (for *B. bjoerkna*) and 0.967 (for *R. rutilus*). In addition, the R^2^ values were >0.90 for *R. rutilus* and *C. auratus* and >0.70 for *B. bjoerkna*. The b values ranged from 2.955 (for *B. bjoerkna*) to 3.265 (for *R. rutilus*). Regarding growth type, positive allometries (b > 3) were found for *R. rutilus* and *C. auratus*, and isometries (b = 3) were found for *B. bjoerkna*.

## 4. Discussion

The Aras Dam reservoir is a unique ecosystem that supports economic and social activities. It provides many benefits for people residing in that area. In addition, many businesses depend on the reservoir, particularly those related to fisheries, aquaculture, local food markets, and small-scale trade. The fisheries in the reservoir are crucial to the economies in numerous cities in north and west Iran, and they also contribute to livelihoods through employment in fishing cooperatives, transportation, and related services. However, little research has been conducted to determine the sustainability and trends of these fisheries, which are difficult to manage. This study is the first to examine the stock status of several fish species in the area by using new data-limited methods and species biometric data.

This study addresses a critical knowledge gap by providing stock status assessments for key species in the Aras Dam reservoir, which were previously unavailable. Using CMSY^++^ combined with species-specific biometric data, our work offers practical insights for fishery managers and policymakers. While this approach improves data confidence, potential biases such as underreporting or species misidentification cannot be ruled out. Future field-based surveys would further strengthen the validation of these estimates. The results of this study can help local fishery managers formulate appropriate fishing strategies for this crucial border ecosystem.

Although stock assessment models are associated with some uncertainties and bias related to overestimation [11,12], they enable rapid stock monitoring. While data on temporal water level changes in the Aras Dam were not available, fluctuations in reservoir hydrology could influence habitat availability, recruitment, and catchability and should be considered in future assessments. Nevertheless, the accuracy of these models may be influenced by specific environmental conditions, the quality of data, and prior parameter settings [25,26]. Because of this, researchers and decision-makers have increasingly used data-limited methods for stock assessment [7,9,10]. The CMSY^++^ model is sensitive to data quality and life-history parameters, but cross-checked catch data and the Bayesian state-space framework help mitigate uncertainties. In addition, research has revealed good agreement between estimates of population parameters obtained using CMSY and those obtained using Schaefer and Fox models [27], as those shown in a stock assessment study using the CMSY model [28]. Thus, the CMSY^++^ method may enable reliable assessment in data-limited fisheries, including assessment of inland stocks such as those in the Aras Dam.

The related BSMs input with additional information (e.g., CPUE data) produce more accurate confidence intervals [29]. In the present study, the CMSY^++^ method was used in conjunction with the BSM to provide a more detailed assessment of the stock status of four species in the Aras Dam reservoir. With the exception of *B. bjoerkna*, which had a healthy status, all species were considerably or moderately depleted, indicating exploitation of these species must be adjusted. This result also indicates that *r*-selected species are capable of resisting strong fishing pressure [10], although lower fecundity may reduce the recovery potential of the stock. Research indicates that adjusting the mesh sizes of gear enables more juveniles to escape capture, which can contribute to stock recovery [10].

For *B. bjoerkna*, this study identified no signs of overfishing or of overfished stock. This indicates that the catch amount for this species can be moderately increased to improve profits and provide economic benefits to fishermen. By contrast, this study indicated that *A. brama* and *C. auratus* stocks were considerably depleted and *R. rutilus* stocks were moderately depleted, indicating these stocks have been overfished. Various measures should be implemented to improve management of the fisheries in the Aras Dam, especially for the species concerned. Studies have recommended several strategies for preserving fish stocks, including imposing total allowable catches, fishing capacity limitations, and size, gear, and season limitations [30,31].

Studies have also reported that climate-driven environmental variability [32,33] and overexploitation [34] play key roles in population fluctuations, and studies have indicated that effective fishery management is the main driver of the recovery of fish stocks [35,36]. Therefore, strict control measures, such as those related to mesh size and season limitations implemented in China [37], especially for illegal fishing, are recommended to achieve balanced stocks and sustainable fisheries. In China, strategies have been implemented to control excessive fishing, strengthening this study’s importance and serving as scientific evidence supporting fishing reduction efforts. In the future, joint international efforts should be expended to manage fishery stocks in the boundary ecosystem of the Aras Dam.

Data on the functional length–weight relationship are crucial to fish stock assessment and management [38]. Length and weight data are required for estimating growth rates and age structure [39] and for calculating standing stock biomass [40], condition indices such as those based on length and weight information [41], and several aspects of fish population dynamics [42]. The best fit values of the parameters for the length–weight relationships were those for *R. rutilus* and *C. auratus* (R^2^ = 0.967 and 0.960, respectively). The results indicate highly significant relationships for *R. rutilus* and *C. auratus* and less significant relationships for *B. bjoerkna* (R^2^ = 0.783). Significant positive allometric growth was noted for *R. rutilus* and *C. auratus*, with a slope of b > 3. This finding indicates that fish become more rotund as the total length increases [43,44].

The findings also reveal isometric growth for *B. bjoerkna*, with b approximating 3. This indicates that the species exhibits identical length and weight growth rates. The values of parameter b in the present study were within the expected range of 2.5–3.5, but they can vary between 2 and 4 [45]. Yılmaz et al. [46] demonstrated positive allometric growth for this species in Lake Ladik, Turkiye. In addition, positive allometric growth has been reported for *R. rutilus* in other regions [47,48].

In general, the length–weight relationship of fish is affected by factors such as season, habitat, gonad maturity, sex, diet, stomach fullness, health, preservation techniques, and annual differences in environmental conditions [45,49]. Differences in b values can be ascribed to one or a combination of factors, including differences in the number of specimens examined, effects of the area/season, the observed length ranges of the specimens caught, and the duration of sample collection [50].

## 5. Conclusions

Different policies can be implemented for border ecosystems such as the Aras Dam. As such an ecosystem, the Aras Dam supports economic and social activities, including fishery and tourism activities. Unmanaged fisheries may lead to a collapse of stocks, and recovery following such a collapse can be time consuming and have social consequences. This study applied the CMSY^++^ model and BSM and considered length–weight relationships to rapidly assess fisheries in the Aras Dam reservoir. The results indicate that the stocks of two species in this reservoir are considerably depleted, and the stocks of the other species are moderately depleted or low. The results were also verified by fishermen through the low fishing activities in the area. To achieve sustainable fisheries in the Aras Dam reservoir, fishery managers can boost stock recovery by adjusting annual catch amounts, implementing mesh size and seasonal limitations, and enforcing strict regulations to control illegal fishing. Our results highlight the need for catch limits aligned with CMSY^++^ reference points and temporary closures during spawning. Declines in fish stocks could also affect local livelihoods, particularly fish powder factories and small-scale trade, so management must balance ecological and economic aspects.

## Figures and Tables

**Figure 1 biology-14-01242-f001:**
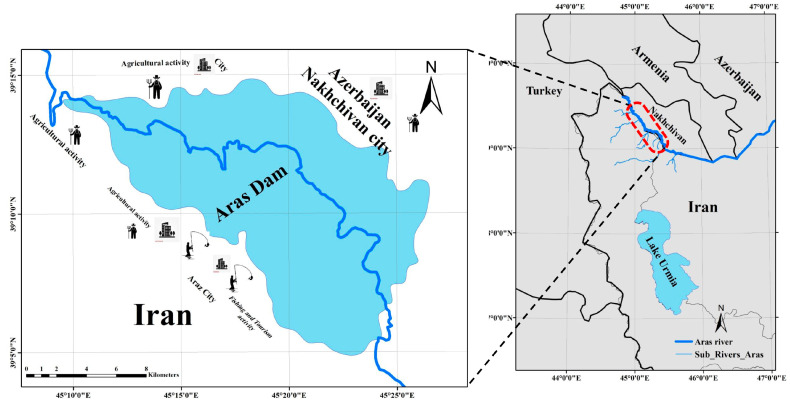
Aras Dam reservoir and fishing locations in northwest Iran.

**Figure 2 biology-14-01242-f002:**
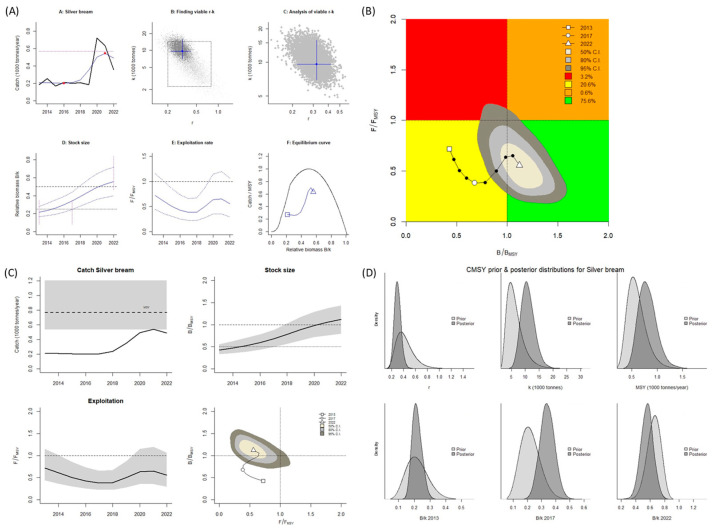
CMSY^++^ model results for *Blicca bjoerkna* in the Aras Dam reservoir from 2013 to 2022. (**A**) shows the time series of catches and smoothed data as used in the estimation of prior biomass by the default rules; (**B**) is a Kobe phase plot; (**C**) is the stock size and exploitation trend; (**D**) the priors (light grey), and posterior understanding (dark grey) of the stock. The colors and symbols are standard use for Kobe phase plot.

**Figure 3 biology-14-01242-f003:**
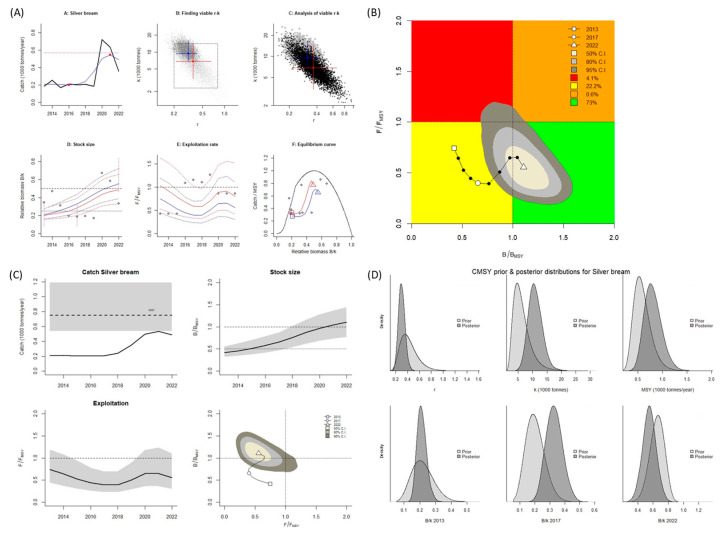
BSM results for *Blicca bjoerkna* in the Aras Dam reservoir from 2013 to 2022. (**A**) shows the time series of catches and smoothed data as used in the estimation of prior biomass by the default rules; (**B**) is a Kobe phase plot; (**C**) is the stock size and exploitation trend; (**D**) the priors (light grey), and posterior understanding (dark grey) of the stock. The colors and symbols are standard use for Kobe phase plot.

**Figure 4 biology-14-01242-f004:**
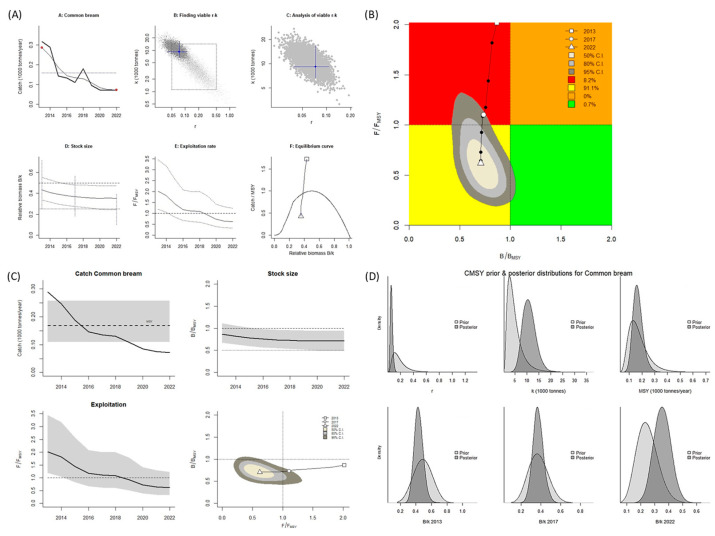
CMSY^++^ model results for *Abramis brama* in the Aras Dam reservoir from 2013 to 2022. (**A**) shows the time series of catches and smoothed data as used in the estimation of prior biomass by the default rules; (**B**) is a Kobe phase plot; (**C**) is the stock size and exploitation trend; (**D**) the priors (light grey), and posterior understanding (dark grey) of the stock. The colors and symbols are standard use for Kobe phase plot.

**Figure 5 biology-14-01242-f005:**
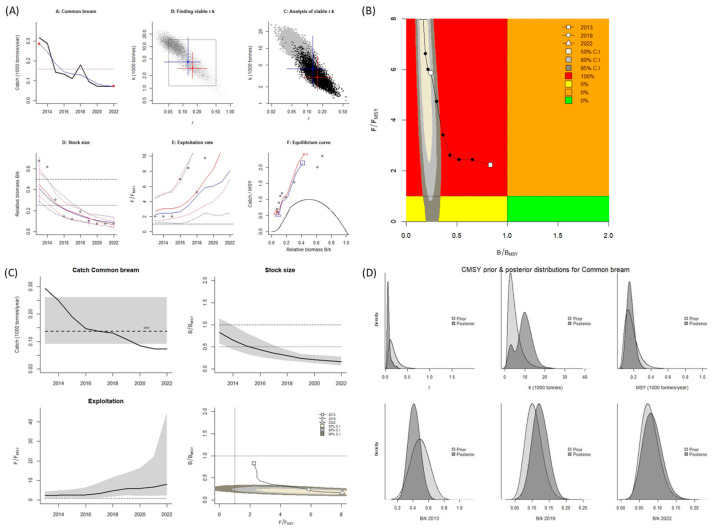
BSM results for *Abramis brama* in the Aras Dam reservoir from 2013 to 2022. (**A**): shows the time series of catches and smoothed data as used in the estimation of prior biomass by the default rules; (**B**): is a Kobe phase plot; (**C**): is the stock size and exploitation trend; (**D**): the priors (light grey), and posterior understanding (dark grey) of the stock. The colors and symbols are standard use for Kobe phase plot.

**Figure 6 biology-14-01242-f006:**
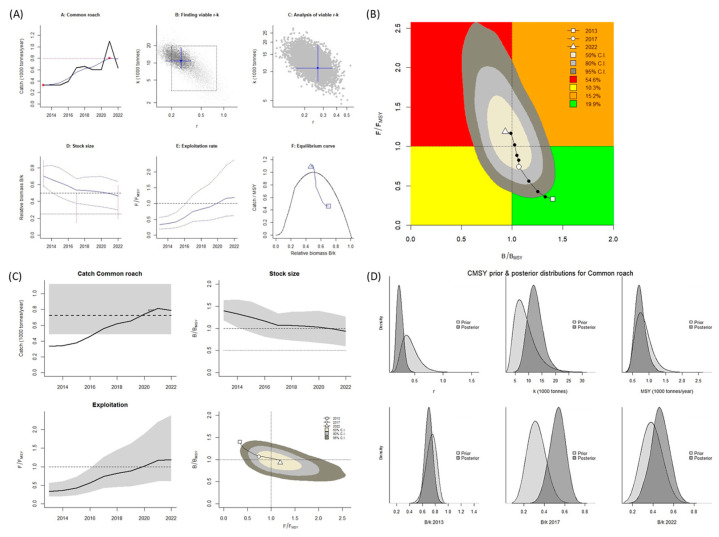
CMSY^++^ model results for *Rutilus rutilus* in the Aras Dam reservoir from 2013 to 2022. (**A**) shows the time series of catches and smoothed data as used in the estimation of prior biomass by the default rules; (**B**) is a Kobe phase plot; (**C**) is the stock size and exploitation trend; (**D**) the priors (light grey), and posterior understanding (dark grey) of the stock. The colors and symbols are standard use for Kobe phase plot.

**Figure 7 biology-14-01242-f007:**
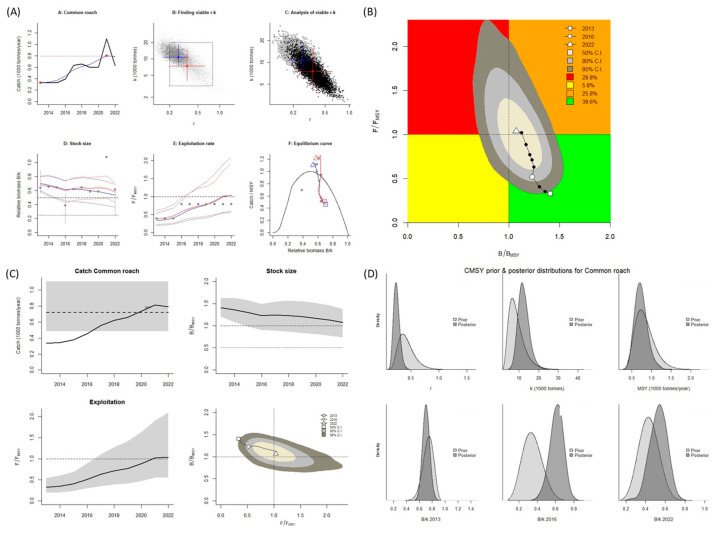
BSM results for *Rutilus rutilus* in the Aras Dam reservoir from 2013 to 2022. (**A**) shows the time series of catches and smoothed data as used in the estimation of prior biomass by the default rules; (**B**) is a Kobe phase plot; (**C**) is the stock size and exploitation trend; (**D**) the priors (light grey), and posterior understanding (dark grey) of the stock. The colors and symbols are standard use for Kobe phase plot.

**Figure 8 biology-14-01242-f008:**
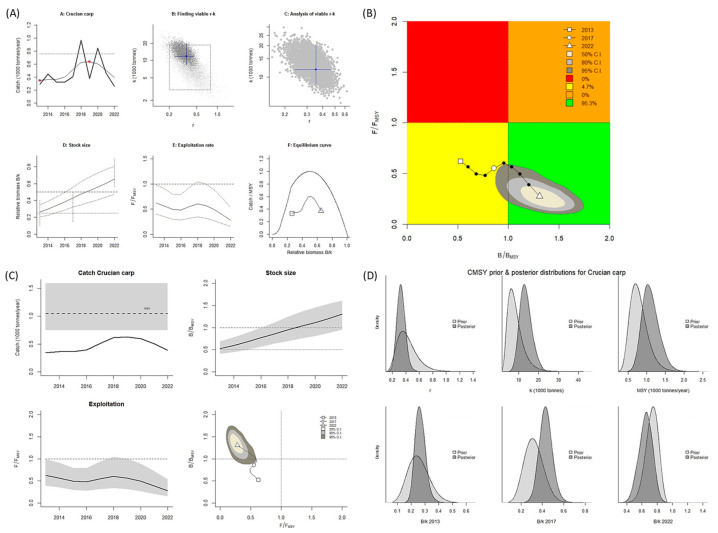
CMSY^++^ model results for *Carassius auratus* in the Aras Dam reservoir from 2013 to 2022. (**A**) shows the time series of catches and smoothed data as used in the estimation of prior biomass by the default rules; (**B**) is a Kobe phase plot; (**C**) is the stock size and exploitation trend; (**D**) the priors (light grey), and posterior understanding (dark grey) of the stock. The colors and symbols are standard use for Kobe phase plot.

**Figure 9 biology-14-01242-f009:**
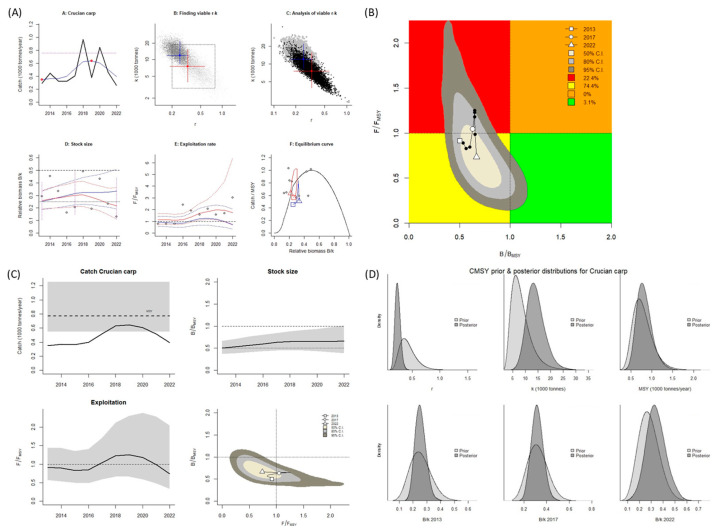
BSM results for *Carassius auratus* in the Aras Dam reservoir from 2013 to 2022. (**A**) shows the time series of catches and smoothed data as used in the estimation of prior biomass by the default rules; (**B**) is a Kobe phase plot; (**C**) is the stock size and exploitation trend; (**D**) the priors (light grey), and posterior understanding (dark grey) of the stock. The colors and symbols are standard use for Kobe phase plot.

**Figure 10 biology-14-01242-f010:**
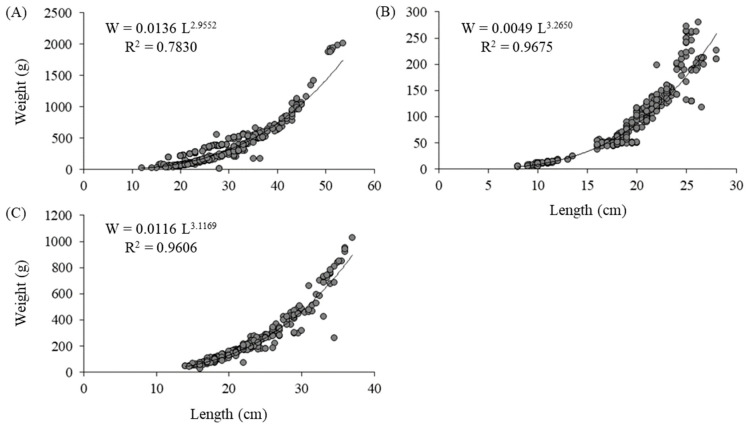
Length–weight (L-W) relationships of noncommercial species in the Aras Dam reservoir for (**A**) *Blicca bjoerkna*, (**B**) *Rutilus rutilus*, and (**C**) *Carassius auratus.* The coefficient of determination (R^2^) is shown in each panel.

**Table 1 biology-14-01242-t001:** CMSY^++^ model and BSM results for predicting key reference point indicators for fisheries of *Blicca bjoerkna* in the Aras Dam reservoir (values in parenthesis represent 2.5th and 97.5th percentiles). Indicators are based on 1000 metric tons (t).

Parameter	Estimated Values	
	CMSY^++^	BSM
Biomass (*B*)	5.98 (3.59–9.63)	5.91 (3.45–9.38)
MSY	0.769 (0.537–1.21)	0.75 (0.537–1.19)
*B_MSY_*	4.82 (3.47–7.96)	4.73 (3.46–7.96)
*F_MSY_*	0.16 (0.11–0.209)	0.159 (0.11–0.207)
*B*/*B_MSY_*	1.12 (0.795–1.44)	1.11 (0.772–1.44)
Exploitation *F*/*F_MSY_*	0.557 (0.294–1.07)	0.556 (0.301–1.1)
Fishing mortality (*F*)	0.0821 (0.0464–0.146)	0.0825 (0.0474–0.153)
Relative biomass in last year	0.559 k (0.397–0.718)	0.553 k (0.386–0.72)
*K*	9.64 (6.93–15.9)	9.46 (6.91–15.9)
*R*	0.319 (0.22–0.418)	0.317 (0.219–0.414)
*B_MSY_*/*K*	0.62	0.62
Depletion level	Low	Low
Trophic level	3.2	

**Table 2 biology-14-01242-t002:** CMSY^++^ model and BSM results for predicting key reference point indicators for fisheries of *Abramis brama* in the Aras Dam reservoir (values in parenthesis represent 2.5th and 97.5th percentiles). Indicators are based on 1000 metric tons (t).

Parameter	Estimated Values	
	CMSY^++^	BSM
Biomass (*B*)	3.76 (2.05–6.38)	0.817 (0.109–1.89)
MSY	0.168 (0.109–0.258)	0.137 (0.0906–0.261)
*B _MSY_*	4.59 (3.28–8.26)	2.13 (1.03–8.33)
*F _MSY_*	0.0365 (0.0196–0.0534)	0.0644 (0.02–0.116)
*B*/*B_MSY_*	0.708 (0.498–0.945)	0.167 (0.083–0.277)
Exploitation *F*/*F_MSY_*	0.62 (0.325–1.23)	8.05 (2.48–45.2)
Fishing mortality (*F*)	0.0197 (0.104–0.0373)	0.0887 (0.0367–0.671)
Relative biomass in last year	0.354 k (0.244–0.473)	0.0836 k (0.0415–0.139)
*K*	9.17 (6.55–16.5)	4.25 (2.07–16.7)
*r*	0.0731 (0.0392–0.107)	0.129 (0.0399–0.232)
*B_MSY_*/*K*	0.41	0.19
Depletion level	Medium	Very Strong
Trophic level	3.1	

## Data Availability

Data from this research will be available upon request to the authors.

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
