# Peer review of "Stock Status of Noncommercial Fish Species in Aras Dam Reservoir: Mismanagement Endangers Sustainable Fisheries"

_biology, 2025, doi:10.3390/biology14091242_

Round 1
Reviewer 1 Report
Comments and Suggestions for Authors
This valuable manuscript applies data-limited assessment methods to a critical freshwater fishery. The findings are robust and communicated clearly. With the incorporation of the suggested revisions, particularly the addition of ecological context and specific management advice, this paper will make a strong contribution to the literature. I recommend acceptance following minor revisions.
We thank the authors for their interesting work and look forward to receiving the revised manuscript.

Author Response
We sincerely thank Reviewer 1 for the thorough and constructive evaluation of our manuscript. We greatly appreciate the positive comments regarding the relevance of our study and the clarity of presentation. Below, we provided detailed responses to the comments and describe the revisions made. These revisions have improved the clarity, ecological relevance, and practical applicability of our study. We sincerely thank the reviewer for their valuable suggestions.
Comments 1: Figures 2–9 The Kobe plots and time-series graphs are informative, but they lack captions that fully interpret the results. Please add brief explanatory notes to the figure captions.
Response 1: Brief explanatory notes have been added to the figure captions to interpret the results more clearly.
A: shows the time series of catches and smoothed data as used in the estimation of prior biomass by the default rules; B: is a Kobe phase plot; C: is the stock size and exploitation trend; D: the priors (light grey), and posterior understanding (dark grey) of the stock.
Comments 2: Tables 1–4 (Units inconsistency) Units are presented inconsistently (Example: 'mt' - 'tons'). Standardize to metric tons (t) throughout
Response 2: All units have been standardized to metric tons (t) throughout the manuscript.
Comments 3: Line 271 (‘B. bipercana’)
Response 3: The species name has been corrected to B. bjoerkna.
Comments 4: References
Response 4: All references have been carefully checked and revised to adhere to MDPI formatting guidelines.
Comments 5: Limnological context
Response 5: Appropriate information has been added in the related sections.
The four species selected account for over 90% of the total catches in the Aras Dam reservoir. Although they are mostly non-commercial, their dominance in terms of abundance and biomass makes them representative indicators for assessing the overall stock status of the reservoir.
Comments 6: Sampling methodology
Response 6: The samples collected by fisherman in a predetermined fixed position using Beach seins net fishing (950-m length, 18-m height, and 35-mm mesh size in the central part of the net). This fixed location determined by expert fisherman where fish are dominant. Fishing period is usually from October to March. We have added appropriate information defined in line 247-250.
including 463 Silver bream, 391 Common roach and 352 Freshwater bream, collected from fishing cooperatives position using Beach seins net fishing (950-m length, 18-m height, and 35-mm mesh size in the central part of the net) once a week for 6 months (October 2023–March 2024).
Comments 7: Illegal fishing estimation
Response 7: Thank you for the comment. Factory records were cross-checked against fisheries landing reports and independent market surveys. We acknowledge potential biases, such as underreporting or species misclassification, and have added a discussion of these limitations in the revised manuscript, line 401-408
While this approach improves data confidence, potential biases such as underreporting or species misidentification cannot be ruled out. Future field-based surveys would further strengthen the validation of these estimates.
Comments 8: Model assumptions and limitations
Response 8: We slightly add sentence in discussion to address this issue. We acknowledge that CMSY⁺ assumes constant carrying capacity and does not account for environmental variability such as water level fluctuations. Although data on temporal changes in the Aras Dam reservoir were not available, we note that such hydrological changes could affect habitat, recruitment, and catchability. Future studies should consider incorporating water level dynamics to improve assessment accuracy.
Estimates of illegal catches derived from fish supplied to factory were cross-checked against official fisheries landing reports and independent local market surveys.
To ensure consistency in interpretation, stock status categories were defined following Froese et al. [22].
Comments 9: Conservation implications
Response 9: We have added sentences already in the Conclusion line 477-487.
To achieve sustainable fisheries in the Aras Dam reservoir, fishery managers can boost stock recovery by adjusting annual catch amounts, implementing mesh size and seasonal limitations, and enforcing strict regulations to control illegal fishing. Our results highlight the need for catch limits aligned with CMSY++ reference points and temporary closures dur-ing spawning. Declines in fish stocks could also affect local livelihoods, particularly fish powder factories and small-scale trade, so management must balance ecological and eco-nomic aspects.
Reviewer 2 Report
Comments and Suggestions for Authors
Review for the paper “Stock status of noncommercial fish species in Aras Dam reservoir: Mismanagement endangers sustainable fisheries” by Ali Haghi Vayghan and co-authors submitted to “Biology”.
The authors of this research paper conducted an analysis focusing on the Aras Dam reservoir, a pivotal inland fishery in northwest Iran, where socioecological pressures threaten sustainable resource use. The study centers on four noncommercial species that are routinely harvested in the reservoir: silver bream, common bream, common roach, and freshwater bream. The results indicate unsustainable conditions for three of the assessed species (common bream, common roach, and freshwater bream) highlighting a pattern of declining biomass or excessive exploitation relative to sustainable benchmarks. In contrast, silver bream exhibits a healthy status, suggesting more stable biomass and a lower risk of immediate depletion for this species under current conditions. Overall, the study contributes to the growing appreciation for rapid, data-constrained assessment tools in inland fishery governance and reinforces the need for adaptive management that responds to species-level differences in stock status.
Recommendations.
Summary
L 18. (Carassius auratus) should be deleted.
Introduction.
L 63-65. The authors' statement that CMSY is "one of the most rapid means" to assess data-poor fisheries requires clarification. What other rapid assessment tools were considered, and what criteria were used to declare CMSY the most suitable for this case?
L 67. Change "Turkey" to "Turkiye"
L 77-79. The authors should report the representativeness of the four noncommercial species chosen (silver bream, common bream, common roach, freshwater bream) for stock-status inference of the reservoir as a whole.
L 80-84. The authors should explain in more detail why this study fills a critical knowledge gap. How does this work advance previous research or address practical challenges faced by the fishery industries? Additional references to prior work and specific implications for managers or policymakers would strengthen the case.
Material and Methods.
The low font size makes Figure 1 difficult to understand.
L 100. The authors' statement that illegal fishing activity was estimated using fish input data of fish powder factories requires clarification. What specific proxies or indicators were used? How were they validated against on-the-ground illegal-fishing evidence?
L 229. The authors should clarify the extent to which CMSY results are sensitive to data quality, input assumptions, and species life-history parameters, particularly in an inland reservoir with mixed commercial and noncommercial fisheries.
L 231. How many specimens of each of the four species were measured? Were there any seasonal trends in sample distribution? The authors should clarify how the time frame of data collection (2013–2022 for CPUE and October 2023–March 2024 for length-weight measurements) affects the comparability of stock status, given potential seasonal and interannual variability.
Results.
L 256. The authors stated, "Overall, no significant difference was noted in the estimates obtained using the two models". What statistical method was used to reach this conclusion?
The authors should increase the font size in figures 2-9.
L 342-346. This text should be moved to the Materials and methods section.
Discussion.
L 356-359. The authors should explain what specific activities and economic sectors are most dependent on the reservoir.
L 379. The authors' claim that “most species were considerably or moderately depleted” needs clarification. Which thresholds or criteria define “considerably” and “moderately” depleted.
L 417. Change "Turkey" to "Turkiye".
The authors should make recommendations for managing these four fish species in the region and discuss how changes in fish stocks could affect local communities that depend on fish powder factories.
Author Response
We sincerely thank Reviewer 2 for the careful and constructive evaluation of our manuscript. The detailed feedback helped improve clarity, methodological transparency, and practical relevance for the manuscript. We provided point-by-point responses below to the comments and outline the revisions made by the reviewer. These revisions improve clarity, rigor, and applicability to the paper. We sincerely thank the reviewer for the insightful suggestions.
Comments 1: L18 Carassius auratus
Response 1: The description mentioned of Carassius auratus has been removed.
Comments 2: L63-65 CMSY as “one of the most rapid means”
Response 2: We slightly revised the sentence.
Nevertheless, CMSY is one of the most rapid methods
Comments 3: L67 “Turkey” → “Turkiye”
Response 3: We have corrected it throughout the manuscript.
Comments 4: L77-79 Representativeness of four noncommercial species
Response 4: Thank you for the comment. We have added a clear rationale in the Introduction section for selecting these species as representative indicators of the Aras reservoir fishery. The four species selected represent the majority of fish captured in the Aras Dam reservoir, with over 90% of total catches attributed to non-commercial species. Although they are not all commercially targeted, these species dominate the fishery in terms of abundance and biomass, making them suitable indicators for assessing the overall stock status of the reservoir. We have clarified this point in the revised manuscript to highlight the representativeness of the selected species.
The four species selected account for over 90% of the total catches in the Aras Dam reservoir. Although they are mostly non-commercial, their dominance in terms of abundance and biomass makes them representative indicators for assessing the overall stock status of the reservoir.
Comments 5: L80-84 Critical knowledge gap
Response 5: Thank you for the comment. This study addresses a critical knowledge gap because little research has previously been conducted on the stock status of fish species in the Aras Dam reservoir, particularly using data-limited methods. By applying CMSY++ in conjunction with species biometric data, our work provides timely and reliable estimates of stock status for both commercial and non-commercial species, which were previously unavailable. These results advance previous research by integrating rapid assessment tools with local species-specific data, offering practical insights for fishery managers and policymakers. For example, our findings can inform decisions on catch limits, gear restrictions, and seasonal regulations, contributing to sustainable management of inland fisheries. Additional sentence has been incorporated into the revised manuscript in discussion line 399-403.
This study is the first to examine the stock status of several fish species in the area by using new data-limited methods and species biometric data. This study addresses a critical knowledge gap by providing stock status assessments for key species in the Aras Dam res-ervoir, which were previously unavailable. Using CMSY++ combined with species-specific biometric data, our work offers practical insights for fishery managers and policymakers.
Comments 6: Figure 1 font size
Response 6: We have improved the quality for the publication need.
Comments 7: L100 Illegal fishing estimation
Response 7: Thank you for the comment. Factory records were cross-checked against fisheries landing reports and independent market surveys. We acknowledge potential biases, such as underreporting or species misclassification, and have added a discussion of these limitations in the revised manuscript.
Comments 8: L229 CMSY sensitivity
Response 8: Thank you for the comment. The CMSY++ model, like any data-limited stock assessment method, is sensitive to data quality, input assumptions, and species life-history parameters. In this study, we used time-series catch and CPUE data collected from legal beach seine fisheries and cross-checked illegal catch estimates against official landing reports and independent market surveys to improve data reliability. Species-specific life-history parameters, such as intrinsic growth rate (r) and resilience, were derived from FishBase and SeaLifeBase, and broad priors for biomass relative to carrying capacity (B/k) were incorporated at the start, mid-year, and end of the time series. The Bayesian state-space implementation in CMSY++ accounts for observation and process error, which allows the model to partially accommodate uncertainties in catch data and life-history assumptions. While these measures reduce sensitivity to data limitations, we acknowledge that variations in data quality, misreporting, or inaccurate life-history inputs can influence stock status estimates. Future field surveys and improved species-specific data would further enhance the robustness of CMSY++ assessments for mixed commercial and non-commercial inland fisheries like the Aras Dam.
Comments 9: L231 Sample size and seasonal trends
Response 9: The samples collected by fisherman in a predetermined fixed position using Beach seins net fishing (950-m length, 18-m height, and 35-mm mesh size in the central part of the net). This fixed location determined by expert fisherman where fish are dominant. Fishing period is usually from October to March. We have added appropriate information as defined in line 237-239.
including 463 Silver bream, 391 Common roach and 352 Freshwater bream, collected from fishing cooperatives position using Beach seins net fishing (950-m length, 18-m height, and 35-mm mesh size in the central part of the net) once a week for 6 months (October 2023–March 2024).
Comments 10: L256 Statistical method for model comparison; Figures 2-9 font size
Response 10: We appreciate the reviewer’s observation. We have removed this statement to avoid potential misinterpretation. Font sizes increased for readability in publication process in a separate file.
Comments 11: L342-346:
Response 11: the text has been moved to the Materials and Methods section.
Comments 12: L356-359 Economic sectors dependent on reservoir
Response 12: the sentence has been revised slightly.
Comments 13: 13. L379 Definition of depletion levels
Response 13: Response: In this study, the classification of stock status followed the criteria suggested by Froese et al. (2023), where stocks with B/BMSY < 0.5 were considered considerably depleted, those with 0.5 ≤ B/BMSY < 0.8 were classified as moderately depleted, and stocks with B/BMSY ≥ 0.8 were regarded as healthy. Appropriate sentence has been added in 2.2 section.
To ensure consistency in interpretation, stock status categories were defined following Froese et al. [22]. Specifically, stocks with B/BMSY < 0.5 were classified as considerably depleted, 0.5 ≤ B/BMSY < 0.8 as moderately depleted, and B/BMSY ≥ 0.8 as healthy.
Comments 14: L417 “Turkey” → “Turkiye”:
Response 14: Corrected the word throughout the manuscript.
Comments 15: Management recommendations and local impacts
Response 15: We thank the reviewer for valuable comments. We have added a section in the Discussion highlighting management recommendations for the four studied species. Specifically, we suggest that (i) stricter monitoring and enforcement of fishing bans be implemented to reduce illegal catches, (ii) seasonal closures be aligned with spawning periods to protect recruitment, and (iii) catch quotas be adjusted based on CMSY++ reference points.
In addition, we emphasize that declines in fish stocks would directly impact local communities, particularly those engaged in fish powder factories, which rely heavily on reservoir landings. Such reductions could lead to loss of income and employment, highlighting the need for sustainable fishery management policies that balance ecological health and socioeconomic stability. Appropriate sentence has been added in conclusion section.
To achieve sustainable fisheries in the Aras Dam reservoir, fishery managers can boost stock recovery by adjusting annual catch amounts, implementing mesh size and seasonal limitations, and enforcing strict regulations to control illegal fishing. Our results highlight the need for catch limits aligned with CMSY++ reference points and temporary closures dur-ing spawning.
Declines in fish stocks could also affect local livelihoods, particularly fish powder factories and small-scale trade, so management must balance ecological and economic aspects.